# Evaluation of a Multi-Objective Genetic Algorithm for Low Impact Development in an Overcrowded City

**Hao-Che Ho** [1][ORCID]**, Shih-Wei Lin** [2]**, Hong-Yuan Lee** [1] **and Cheng-Chia Huang** [1,]***

[1]   Department of Civil Engineering, National Taiwan University, Taipei City 106, Taiwan;
     haocheho@ntu.edu.tw (H.-C.H.); leehy@ntu.edu.tw (H.-Y.L.)
[2]   Sinotech Engineering Consultants, Ltd., Taipei City 105, Taiwan; linswayofficial@gmail.com
[*]   Correspondence: chengchiahuang@gmail.com

**Abstract:** Sustainability and resilience are up-to-date considerations for urban developments in terms of flood mitigation. These considerations usually pose a new challenge to the urban planner because the achievement of a sustainable design through low impact development (LID) practices would be affected by the selection and the distribution of them. This study proposed a means to optimize the distribution of LIDs with the concept of considering the reduction of the flood peak and the hydrologic footprint residence (HFR). The study region is a densely populated place located in New Taipei City. This place has been developing for more than 40 years with completive sewer systems; therefore, the design must consider the space limitations. The flood reduction induced by each LID component under different rainfall return periods was estimated, and then the detention ponds were also conducted to compare the improvements. The results showed that the performance of LIDs dramatically decreased when the return periods were larger than ten years. A multi-objective genetic algorithm (MOGA) was then applied to optimize the spatial distribution of LIDs under different budget scenarios, and to decide the priority of locations for the LID configuration. Finally, the Monte Carlo test was used to test the relationship between the optimal space configuration of LIDs and the impermeability of the study region. A positive correlation was uncovered between the optimal allocation ratio and the impermeable rate of the partition. The study results can provide general guidelines for urban planners to design LIDs in urban areas.

**Keywords:** urban resilience; LID; HFR; MOGA; Monte Carlo test

## 1. Introduction

According to the report from the United Nations, 55% of the seven billion population is concentrated in cities, and urban drainage problems are increasing as a result of the high density population [1]. In the past 20 years the idea of the drainage system design was focused on the rapid transmission of water from the city to downstream, but nowadays the urban rainstorm management has shifted from drainage regulation to source control [2]. The best management practices (BMPs) including detention basins and sediment deposition pools are applied to store the runoff and pollutants, which could achieve flood and pollution reduction. Low impact development (LID), on the other hand, features the installation of small rainwater treatment units to minimize hydrological changes such as runoff, infiltration and evaporation after land development in urban areas. LID units include ecological retention units, rain gardens, green roofs, permeable pavements, rain barrels, and bioswales [3]. Green infrastructure (GI) is the integration of maximum green spaces and corridors into the urban planning to improve urban micro-climate while taking into account the urban ecology [4]. However, a wide range of factors need to be considered for GI. There is no model which can simultaneously simulate surface water, groundwater, the hydrological cycle, and the urban heat island effect. BMPs are often

limited to optimize the flood space in a city, while the optimization of LID distribution seems the efficient method to enhance the resilience of the city.

In the past, the study on the LID design for storm control in urban areas has been simulated with hydrological models [5]. For numerical models about LID simulation in urban areas, it shows that Storm Water Management Model (SWMM), developed by U.S. Environmental Protection Agency, works well under the considerations of time, space, accuracy, and calculation efficiency [6–8]. The flood reduction performance of various LID units shows different effects under different scenarios. Ho [9] points out that the bioretention cell is superior to the permeable pavement, and does not recommend the use of bioswales. Zhang et al. [10] show that the bioretention cell has the best performance for flood reduction, and the performance of the green roof is equivalent to the permeable pavement. Giacomoni et al. [11] present that the green roof is superior to the permeable pavement for the flood reduction rate. Eckart et al. [12] demonstrate that the bioretention cell combined with the bioswale could perform the best effect, but the permeable pavement and the rain bucket do not. The aforementioned results tell that the effectiveness of LID units is limited to the characteristics of local conditions; therefore, the optimization to determine the configuration of LID units in urban areas becomes important. Duan et al. [13] conducted SWMM combined with modified particle swarm optimization (MPSO) to analyze multi-objective optimization designs for detention ponds and LID units. Jung et al. [14] calculate the optimum design of the permeable pavement using the harmonic search method (HS). Liu et al. [15] develop a new model which is named Long-Term Hydrologic Impact Assessment—Low Impact Development 2.1 (L-THIA-LID 2.1) to estimate the optimization of LID configurations to mitigate the impact of urbanization and climate change. Liang [16] apply a genetic algorithm (GA) combined with SWMM to seek the best configuration of campus LID for 36 types of rainfall patterns. Eckart et al. [12] include the Borg multi-objective evolutionary algorithm (Borg Moea) to explore the best combination of LID on small scales under different budgets.

The discussion of the urban storm control has emphasized the reduction of the flood peak, but only represents the hydrologic quantity of a certain point in space. The flood peak is not enough to show the flood situation during storm events. In view of this, Giacomoni et al. [17,18] propose the idea of the hydrologic footprint residence (HFR) to reflect the water transport situation in the open channel drainage during the storm events, and demonstrate that the degree of the land development would affect HFR. They also find that the distribution of LID in cities might affect HFR [11]. Although they propose the idea other than controlling the flood peak, the small-scale study cases cannot fully represent the characteristics of the drainage system in cities. Moreover, the complexities of the urban flood needs a proper index to delineate. LIDs can mitigate the flooding threats for storm events, but they are expensive to install and maintain. With budget considerations and efficiency, it is inevitable to optimize the LID distribution to obtain the greatest hydrological benefits in the city. The reduction of the flood peak with LIDs revealed that the longer the return period of events, the less efficient the LIDs. However, the performance of each LID unit under different return periods has not been examined, especially for urban regions. In this study the performance of each LID unit was explored under 14 scenarios in the metropolitan area, where the highest density population region in New Taipei City is located. The designed scenarios included two durations combined with seven return periods to investigate the efficiency of each LID unit. Multi-objective genetic algorithm (MOGA) and Monte Carlo tests then were applied to evaluate the distribution of LID units in the city under the considerations of flood peak reduction, the HFR and the construction cost.

## 2. Study Region

The study region is located in New Taipei City, covering the administrative regions of Yonghe, Zhonghe, Banqiao, and Tucheng Districts (see Figure 1), with a total area of 24.22 km$^2$ and a population of 630,000 in the northern part of Taiwan. The topography in the southeastern side is higher than the northwestern side. Most of the flooding problems were caused by the intensive rainfall overwhelming the capacity of drainage systems. Therefore, urban flooding during storm events, instead of riverine

flooding, was discussed in this study. There are four major drainages to pass the water into rivers (Figure 1), of which the south side waterway can divert most of the flow from the southern region into Xindian Creek, greatly reducing the flood flow from the mountains into the urban areas; the others have been rehabilitated as concreted artificial channels. The study region was originally planned for a population of 30,000, but with the rapid development it has developed far beyond the planned population, leaving most of the land turned into residential regions. At present, the population density in the study region is about 26,100 people per km², which is the densest area in Taiwan. Most of the green spaces have become residential areas and the width of roads have become narrow (less than six meters). The green coverage ratio, the open space and the designed population have deviated from the original urban planning in this region. The space for re-planning can be severely restricted, and the capacity of the drainage system is no longer suitable for the current condition. Therefore, it is necessary to consider the optimization of LID distribution and combine the existing infrastructure to mitigate the impact of flooding.

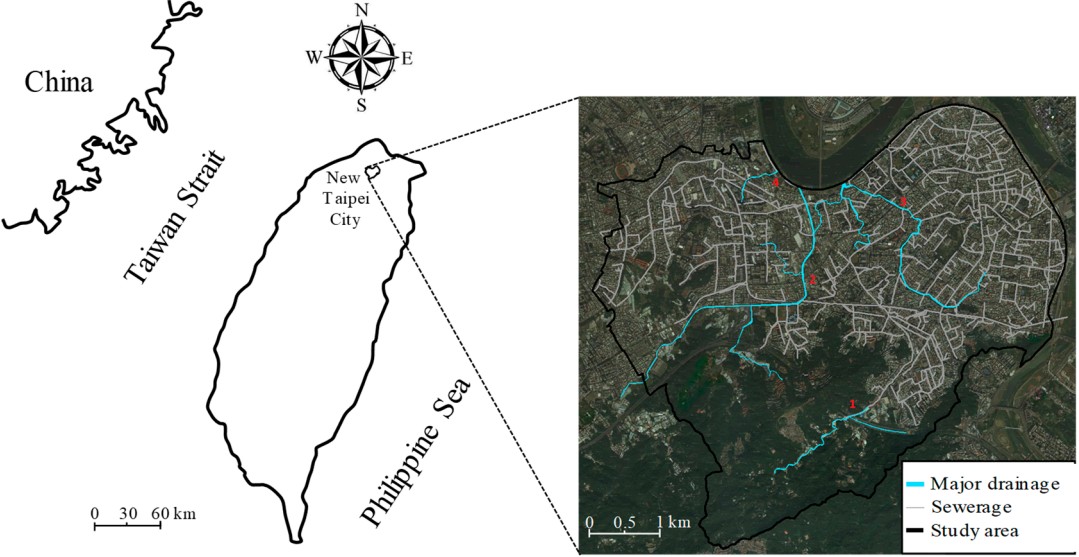

**Figure 1.** Study region and the drainage system.

## 3. Experimental Methods

### 3.1. SWMM Setup

SWMM is usually used for storm event simulation in urban areas. It can delineate the urban rainfall-runoff process regarding water quantity and quality with one-dimensional equations. SWMM version 5.1 was conducted in this study, and the surface runoff and flow routing modules were used to simulate the surface water flowing into the drainage sewer system. All LID components built-in SWMM were conducted to evaluate the performance in 14 scenarios. Each scenario defined with Horner's equation was in accordance with the values in the Hydrological Design Application manual:

$$I_t = \frac{a}{(t + b)^c} \tag{1}$$

where $I$ is rainfall intensity, $t$ is duration, and $a$, $b$, $c$ are parameters calculated with historical field data. There were seven return periods and two durations simulated based on the equation. All parameters used for the equation are shown in Table 1.

**Table 1.** The parameters used for different scenarios for the designed simulations.

| Return Periods | 2-year | 5-year | 10-year | 25-year | 50-year | 100-year | 200-year |
|---|---|---|---|---|---|---|---|
| a | 1767.13 | 2546.37 | 3290.83 | 4741.47 | 6457.74 | 9137.76 | 13,435.59 |
| b | 17.31 | 25.03 | 33.65 | 49.20 | 64.86 | 84.75 | 109.34 |
| c | 0.7679 | 0.7679 | 0.7759 | 0.7969 | 0.8204 | 0.8506 | 0.8872 |
| $R^2$ | 0.9930 | 0.9944 | 0.9966 | 0.9971 | 0.9955 | 0.9928 | 0.9893 |

The sub-catchments were delineated based on the land use data in the study area. There were 20,341 records of land use data. To reduce the computational time, they were reduced to 6639 sub-catchments by merging records with the same land use type that were next to each other. The area of each sub-catchment was calculated with ArcGIS software, and the slope was decided based on the digital elevation model. There were ten types of the land use. The impervious portion, Manning's n for overland flow over the impervious portion (N-Imperv), and Manning's n for the pervious portion (N-Perv) were decided based on the field data and listed in Table 2 (data can be retrieved from http://wrs.ntpc.gov.tw/rwweb/). The depth of depression storage on impervious and pervious portions were set as 10 mm and 15 mm from the SWMM manual.

**Table 2.** The parameters for sub-catchments.

| Land Use Type | Portion (%) | Impervious (%) | N-Imperv | N-Perv |
|---|---|---|---|---|
| Forest | 21.70 | 0 | - | 0.02 |
| Agricultural land | 4.60 | 18 | 0.404 | 0.02 |
| Park | 2.55 | 19 | 0.404 | 0.02 |
| School | 3.38 | 60 | 0.013 | 0.02 |
| Governmental unit | 0.26 | 78 | 0.013 | 0.02 |
| Residential zone | 32.20 | 84 | 0.013 | 0.02 |
| Land for Public facility | 5.87 | 87 | 0.013 | 0.02 |
| Industrial zone | 4.72 | 90 | 0.013 | 0.02 |
| Commercial zone | 3.45 | 97 | 0.013 | 0.02 |
| Road | 21.27 | 100 | 0.013 | - |

There were 3344 junctions and 3355 conduits in this study area. The information related to coordinate, elevation, geometry, and length was downloaded from the field data. The Manning's n for conduits was set as 0.015, and for four major drainages was set as 0.025. The slope and width for major drainages were based on the field data.

In setting up the LID component in sub-catchments, this study took into account the characteristics of the land, and selected the most suitable LID component to be configured, which was defined as the "LID of suitability principle". For example, the residential zone in the central study region had almost no space to place LID components. The green roof was selected to be placed because of its low-cost for installation and maintenance. Moreover, the commercial zone could afford the higher cost of LID components, so rain barrels (or rainwater harvesting) were chosen. Parks and schools had a large number of ecological retention units. The governmental agencies and units might have had more space to contain the less expensive rainwater garden. The parking spaces in the industrial zone could be replaced with permeable pavement. The road and the sidewalks were replaced with permeable pavement. The limitation of LID installation in each sub-catchment was considered for the optimization in this study. The area containing LID was smaller than the area of each zone timing reduction factor:

$$0 \leq A_{LID} \leq \varphi A_S \tag{2}$$

where $A_{LID}$ is the area for the LID component, $\varphi$ is the reduction factor, and $A_S$ is the area of the sub-catchment. The factors are shown in Table 3.

**Table 3.** The low impact development (LID) setting for different land use.

| Land Use | Impervious Percentage | LID Component | $\varphi$ |
|---|---|---|---|
| Residential zone | 60% | Green roof | 0.6 |
| Commercial zone | 95% | Rain barrel | 0.1 |
| Park | 90% | Bioretention cell (bioswales) | 0.5 |
| School | 90% | Bioretention cell (bioswales) | 0.2 |
| Governmental unit | 50% | Rain Garden | 0.2 |
| Industrial zone | 20% | Permeable Pavement | 0.1 |
| Road | 10% | Permeable Pavement | 0.1 |

### 3.2. Hydrological Footprint Residence

Climate change and urbanization change the hydrologic cycle in the watershed. The urban runoff and the flood risk increase because of the expanded impermeable area. During storm events, increasing flood peak and duration can also damage the ecological system in the area. The conventional strategy for storm control emphasizes the reduction of the flood peak at the outlet of the watershed by the evaluation of the land use under different hydrologic conditions. However, the flood peak cannot reflect the impact of the flood toward the watershed. In this study, an index, hydrological footprint residence (HFR), was conducted to quantify the impact of the area during the storm event. The HFR is product of the area of land that is inundated, and the duration over which it is inundated as a storm wave passes through a specified reach of a receiving water body [18]. It is expressed in the unit of area–time. The HFR, in other words, is the term to represent the area under the inundated land time series curve.

The calculation of the HFR takes Figure 2 [18] as an example. The cross-section of the river channel is shown in Figure 2a, and assumes the reach is 100 m. The hydrograph passing the river is shown in Figure 2b. The corresponding water depth can be expressed with Figure 2c. The inundate area for this reach in time series can be estimated (see Figure 2d). The HFR then is the area under the curve of Figure 2d for this rainfall event. The HFR can be regarded as the product of the overflow area of the river and its flooding time. The HFR compared to the flow peak can contain information about the river reach instead of a point information during the event. Therefore, the HFR compared to the flow peak can be the proper index to analyze hydrological changes and use it to develop watershed management plans.

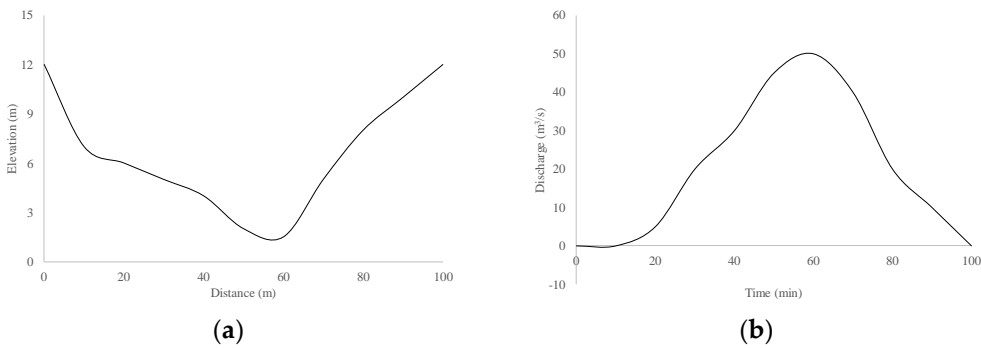

(a)　　　　　　　　　　　　　　　　　　　　　　　(b)

**Figure 2.** *Cont.*

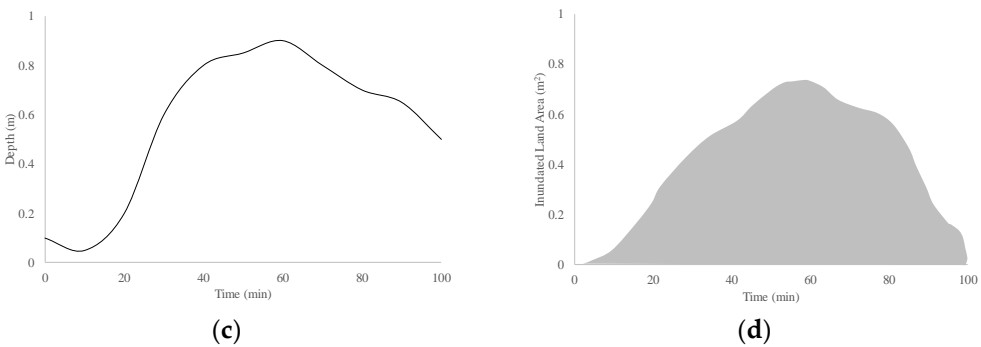

**Figure 2.** Calculation method of hydrologic footprint residence (HFR) (**a**) cross section; (**b**) rainfall-runoff calendar; (**c**) corresponding bathymetric calendar; (**d**) curve of the immersion area of the river section changes. Redrawn by this study [18].

The four major drainages in the study were artificial concrete drainages. The section shape was a trapezoid, and the width of the water surface (*TW*) can be directly calculated from the water depth. The inundated area (*A*) is the product of the width and the length (*L*) of the river. The HFR is the integral over the period of time (*t*) by [17]:

$$A_t = \sum_{i=1}^{N} TW_i \times L_i \tag{3}$$

$$HFR = \sum_{t=0}^{T} A_t \Delta t \tag{4}$$

*3.3. Optimization Approach*

A multi-objective optimization approach was conducted to evaluate the priority areas to set the LID component in the study region. Gene algorithm, also known as a genetic algorithm, is to apply the concept of natural selection to solve the optimization problem [19]. The most significant feature of this method is the degree of freedom, even if it has discontinuous, non-differentiable, stochastic, or highly nonlinear functions. In this study the MOGA was adopted for the optimization problem. The fitness function is the objective function of the optimization problem. These functions of this study were as follows: the variation of the flood peak flow of downstream outlet during a rainstorm, the HFR variation of draining ditch water flow during a rainstorm, and the cost of the configuration of LID. Among them, the lower cost of LID and HFR caused a higher flood peak flow. On the contrary, the greater cost of LID and HFR caused a smaller flood peak flow. Thus, the best configuration of LID and HFR could be achieved when the combination of simultaneous reduction of flood peak and setup costs.

$$min \begin{cases} \Delta HFR = HFR_a - HFR_b \\ cost = \sum_{i}^{N} \sum_{j}^{n} (u_j \times A_{i,j}) \end{cases} \tag{5}$$

$$min \begin{cases} \Delta peak = peak_a - peak_b \\ cost = \sum_{i}^{N} \sum_{j}^{n} (u_j \times A_{i,j}) \end{cases} \tag{6}$$

where, $\Delta HFR$: variation of HFR; $HFR_a$: HFR before the deployment of LID; $HFR_b$: HFR after the deployment of LID; $u_j$: setup cost of unit area of $j_{th}$ LID; $A_{i,j}$: $j_{th}$ LID of $i_{th}$ watershed setup; $\Delta peak$: variation of the peak discharge; $peak_a$: peak discharge before the deployment of LID, $peak_b$: peak discharge after the deployment of LID.

In the problem of multi-objective optimization, the fitness functions are inversely proportional to each other, and it is not possible to find the individual with the lowest fitness. Therefore, the problem does not appear as a single optimal solution. To find the Pareto frontier (illustrated in Figure 3) the cost and flood peak flow were obtained separately under different LID components in this study. By finding these non-dominant solutions in the program, the Pareto frontier was used to delineate the result. In this study, the number of Pareto best solutions was needed to be adjusted and the geometric average propagation value of 50 consecutive generations was set at less than 0.0001 for the final Pareto optimal solution.

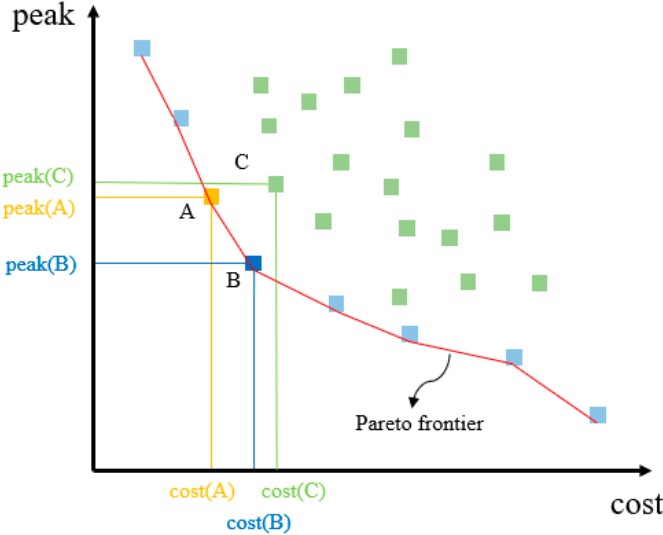

**Figure 3.** Pareto frontier of multi-objective optimization.

## 4. Results and Discussions

The parameters for the SWMM simulations were verified with inundated places in the study area. The SWMM was used for flow routing in storm sewers and the two-dimensional overland flow model was employed for the calculation of surface flooding [20]. The inlets and junctions of the sewer systems were applied as the connecting points between SWMM and the two-dimensional overland flow model. The inlet was treated as the sink and the junctions were treated as source for this two-dimensional model. The detail of the interaction simulation between SWMM and two-dimensional overland flow model can be referred to in the previous study [20]. The inundated places simulated in this study were compared to the surveyed data in two typhoon events (Typhoon Herb and Typhoon Nari). The accuracy was 96.3% and 89.5%; therefore the settings employed in SWMM were suitable for the LID simulation.

### 4.1. Influence of Different Rainfall Durations on LID and Detention Pond

The performance of LIDs to reduce the flood peak and the HFR were investigated under 14 scenarios. For the six-hour duration case, the flood reduction by LIDs could reach the maximum volume for the two-year return period, and became constant after the 10-year return period. The reduction by detention ponds, on the other hand, could increase with the return periods (see Figure 4a). This result told us that the detention pond could make up the disadvantage of the LID for large storm events, and the combination of them could be the proper strategy for the flood mitigation. Bai et al. [21] also suggests that a proper design to combine LIDs and detention ponds might be the solution for storm management in the urban area. For the 24-hour duration case, the performance of LIDs to reduce the flood peak generally was poor when compared to the detention ponds. This showed that the capability of LIDs to reduce the floods was low for long durations. For the long duration and intensive rainfall cases, the detention ponds were more suitable than LIDs for flood reduction. The decreasing

trend could be observed for the performance of LIDs to reduce the flood peak and the limitation was the 10-year return period for the six-hour duration case (see Figure 4b). In terms of the 24-hour duration, the performance of the detention ponds kept the reduction rate at least 3% throughout all return periods. The above results demonstrated that LIDs can only reduce the flood peak effectively in the short duration and low intensity rainfall; moreover, they could play the advantage if the return period was less than 10 years. Pereira Souza et al. [22] conduct a hydrological modeling for the performance of LID and show that the 10-year return period rainfall is the limitation for storm control in urban areas. The function of LIDs essentially was increasing infiltration and containing the runoff by increasing many small storage spaces. The infiltration rate would be reduced over time, and once the rainfall duration was long and reached the limitation of infiltration rate, the effect of LID units would be minimized. Therefore, if the vulnerability of cities is caused by the short duration with the low return period, LIDs should be first considered, otherwise detention ponds should be given the priority. Rezaei et al. [23] also demonstrates that detention ponds are the proper solution to the flood control for long durations. In this study, the effect of detention ponds for flood reduction did not perform very well because those ponds were allocated in the existing parks and schools of the study area and these spaces were already designed at the end of the drainage system. In fact, the performance of the dentation ponds was highly affected by the location. If ponds were allocated at the downstream of the drainage system, the reduction rate would be less than at the upstream of the system. Those ponds were only allocated in the existing parks and schools since the study area was crowded and developed, and there was no available space for retrofitting. Even though the performance of detention ponds was not the best for this simulation, the trend of results showed that the detention ponds were superior than LIDs in the large return periods.

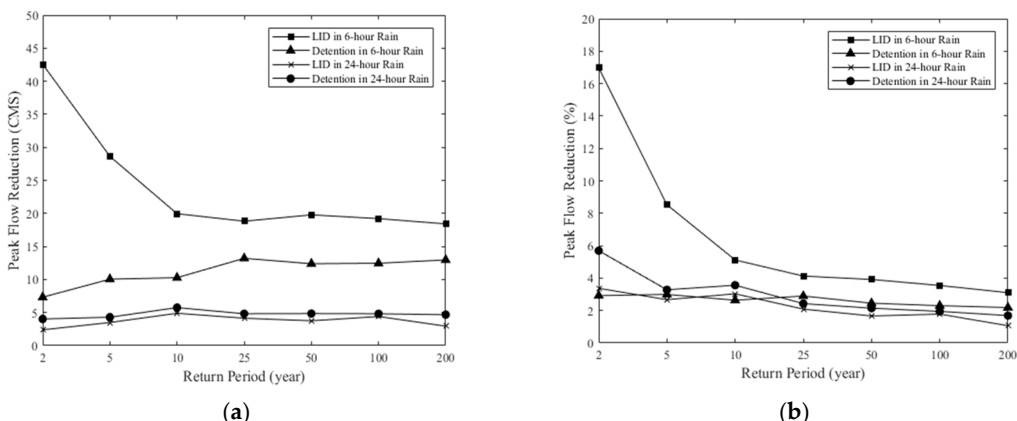

**Figure 4.** The flood peak reduction by LIDs and detention ponds for different rainfall durations and return periods: (**a**) quantify of the peak flow reduction; (**b**) percentage of the peak flow reduction.

The HFR could be reduced by LIDs with increasing return periods for six-hour duration, but the amount of variation was not significant (Figure 5). The HFR reduced by detention ponds roughly increased with the return periods. There was no significant change for different periods, but the overall performance was far less than LIDs. For six-hour duration, there was no particular relationship between the HFR reduction and the return periods. For 24-hour duration, LIDs reduced the HFR with 70% of the amount of the six-hour duration. The performance of detention ponds was low for the HFR reduction. Overall, the reduction rate and the reduction amount for different periods were mostly similar. The detention ponds could efficiently reduce the flood peak but not for the HFR. The main reason for which was the different mechanism of the storm control. The detention pond accumulated the surface runoff during the flood peak in a certain place and released it to the downstream after the flood peak, which would have an adverse effect on the downstream. In other words, the HFR could be reduced before the flood peak but be increased after it. The overall HFR reduction was small. LIDs increased infiltration and fundamentally reduced the surface runoff; therefore, the runoff could

be reduced throughout the event and the overall HFR could also be reduced. LIDs comparing to detention ponds were proper practices to mitigate the impact toward the downstream.

The aforementioned results present that LIDs could control the storm event under the short duration better than the long duration in the urban area. Those simulations were done based on the adaptive distribution of LIDs and all components were conducted. For LID practices, it was necessary to evaluate the performance of each component, and select the best efficiency one as the priority. In order to explore the performance of the flood reduction for each component, every LID component was simulated individually, and was fully distributed to the available spaces within the region. The performance of each component was judged by the reduction rate per component area and per component cost. Figure 6 shows the results for the six-hour duration. The rain barrel had the highest reduction rate per area, but after a 10-year return period it did not show an obvious advantage (see Figure 6a). The other LID components were far less effective than the rain barrel. In terms of the reduction rate per cost, the green roof and the permeable pavement performed better than others (see Figure 6b). Before the 10-year return period, the green roof performed better than the permeable pavement, but after that the permeable pavement was better. It also showed that the permeable pavement was less affected by the return periods. Liang et al. [24] indicate that the selection of one LID component over the others highly depends on the return period and the duration. If the unit area and the unit cost were considered at the same time, the rain barrel could simultaneously reach the lesser area and cost but a larger flood reduction rate. The green roof had the low construction cost, but it needed a lot of space to reach the effective flood peak reduction. For the HFR reduction rate of each component per area, it showed that regardless of the return periods, the rain barrel still maximized the benefit compared to others and even performed better in high return periods (see Figure 7a). The permeable pavement was second but the effect of reducing the HFR per area decreased with the increase of the return period. In terms of the HFR reduction rate per cost, the permeable pavement was the most advantageous regardless of the return period (See Figure 7b). The rain barrel performed better in the low return period than the green roof, the rain garden, and the bioswale, but the green roof performed better in the high return period.

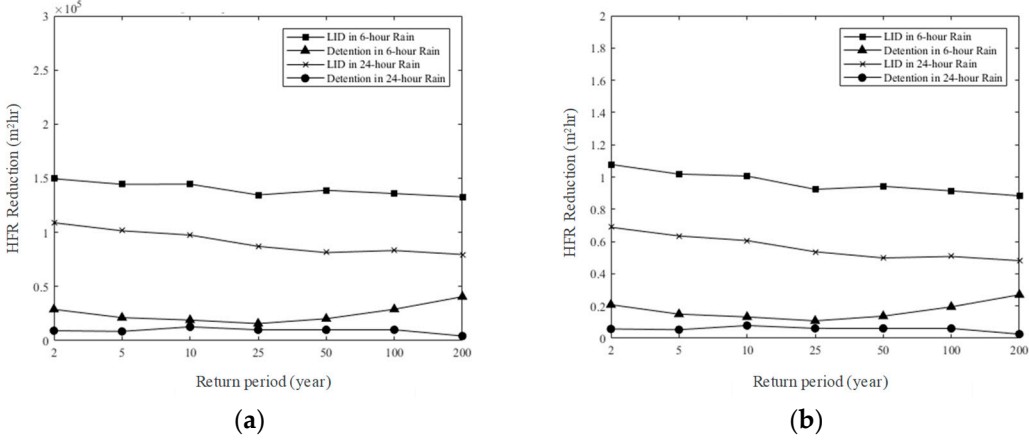

**Figure 5.** The HFR reduction by LIDs and detention ponds on different rainfall durations and return periods: (**a**) quantify of the HFR reduction; (**b**) percentage of the HFR reduction.

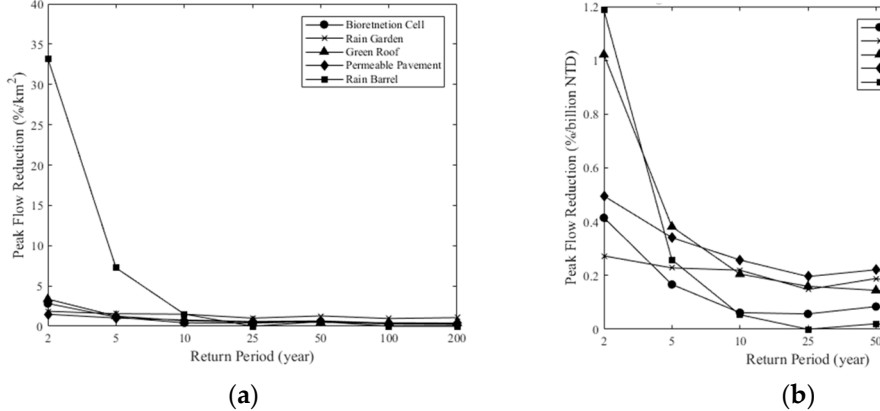

**Figure 6.** The flood peak reduction by different LID components for the six-hour rainfall duration and different return periods: (**a**) percentage of the peak flow reduction for unit area in 6-hour rainfall; (**b**) percentage of the peak flow reduction for unit cost in 6-hour rainfall.

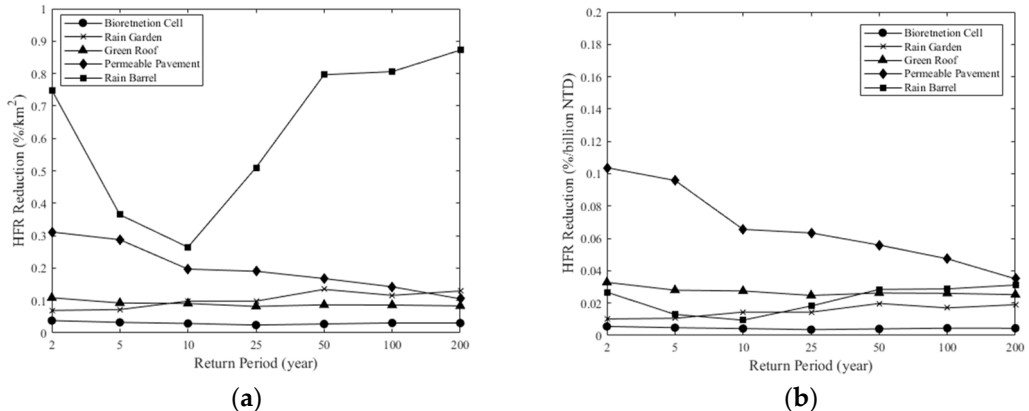

**Figure 7.** The HFR reduction by different LID components for the six-hour rainfall duration and different return periods: (**a**) percentage of the HFR reduction for unit area in 6-hour rainfall; (**b**) percentage of the HFR reduction for unit cost in 6-hour rainfall.

If the maximum efficiency to reduce the flood peak and the HFR in a limited area is the target, the rain barrel should be the best choice and then followed by the permeable pavement. If the cost is the main concern, the priority should be given to the permeable pavement and then followed by the green roof and the rain barrel. The rain barrel can obtain the large surface runoff per component area, but it is expensive. On the other hand, the permeable pavement and the green roof can obtain their own areas of the surface runoff, and replace the land which has a high impermeable rate, but the cost is inexpensive. In the simulation cases they performed second compared to the rain barrel. The bioswale could only be installed in places with low impermeable rates, and these places had less surface runoff. If the benefits of each LID component were compared under different conditions, it could be found that the proper LID component could perform much better than others under the low return period. This indicated the importance of the optimization for the LID distribution.

*4.2. Optimization Method of LID Distribution*

Installation of a large number of LIDs in the city is very costly. The prioritization of LIDs is important to ensure the maximum efficiency during storm events. The cost usually will be proportional to the reduction of the water volume. Therefore, this study showed how to achieve the maximum flood reduction through the LID optimization in highly developed regions without changing the original land use design. The MOGA was used to explore the prioritization of LIDs under different costs to reach the maximum reduction of the flood peak and the HFR. The existing administrative partitions of the study region were complicated and fragmented (there were 163 in total), therefore we considered

the geographical location, the population size, and the simulation time, and then reorganized them into 53 partitions. We ensured that each partition contained 10,000 people, and the optimization would calculate the ratio of area including LIDs for each partition. The ratio was defined as the area including LIDs dividing the available space. In order to enhance the simulation efficiency and avoid the fragmented configuration, we started with the ratio, allocated LIDs in this partition, and then performed the optimization. The updated ratio was then applied in the new iteration. The calculation was continued until the optimal ratio was reached.

The effect of different intensity rainfall on LID was analyzed and showed that using LIDs could successfully reduce the flood peak for the low return period. The scenario was then designed with six-hour duration and five-year return period to assess the LID optimal ratio of each partition (Figure 8). The five-year return period is usually the designed criteria for urban drainage systems. Detention ponds were usually limited by the space and the selection was also limited, and LIDs were flexible and could apply appropriate components according to the different land uses, therefore, only the spatial configuration of the LID components was discussed, regardless of the configuration of the detention ponds. In this study, MOGA function "gamultiobj" in MATLAB r2017a was conducted to do optimization. The setting variable was the ratio of area including LIDs for each partition, and the objective function was the reduction of the flood peak and the LID cost, or the reduction of the HFR and the LID cost. The flood peak flow data was obtained directly by SWMM simulation, while the HFR was calculated indirectly by SWMM simulation of the water depth sequence of the major drainages. The cost of LID configuration was estimated based on the LID manual. The remaining parameters, such as the number of contemporary individuals was 200, the best individual rate was 0.15, the use of the single-point mating method, the mating rate was 0.8, and the mutation rate was 0.05. Finally, when the average propagation value was less than 0.0001 or the total generation was greater than 2000, the calculation was stopped and five times the evaluation was calculated to check the result sufficiently converged.

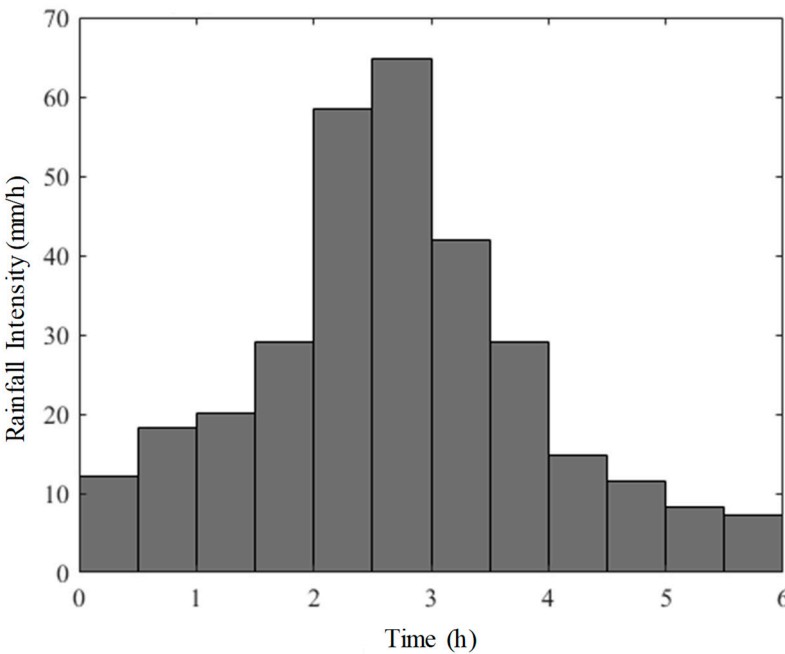

**Figure 8.** Design hyetograph for 6-hour duration of 5-year return period.

*4.3. The Spatial Configuration of the LID from Optimization*

The previous sections showed that the performance of LIDs would be affected by the rainfall intensity and the duration. The configuration of LID in different locations was investigated to learn the impacts on the reduction. If we allowed the maximum area to place a specific LID component,

calculated the reduction of the flood peak or the HFR, and then divided the cost for each partition in the study region, we could simply decide the priority based on this result without the optimization. However, the different combinations of the ratio of LIDs in each partition would have interacted effects for the reduction. In addition, there was no means to decide how many LIDs should be set in partitions of the region under the specific budget. Therefore, in this study the different ratios of the LID configuration and the different LID components in each partition were calculated with the MOGA method to obtain the optimized LID distribution for the region under the specific budget.

Without setting LID in the study region the flood peak flow was 176 m³/s and this was defined as 100% of the peak flow. If we assumed each partition could contain the maximum area of the LID, the total cost would be $1.1 billion which defined as 100% of the LID cost (the cost was estimated based on the LID manual). After the MOGA simulation, the Pareto frontier between the peak flow and the cost was obtained (see Figure 9). It showed that when the cost of the LID was more than 80%, the flood peak flow was almost the same as the 100% cost. This meant that with the LIDs setting up to 80% the flood reduction might reach the maximum for this region. In terms of optimal spatial configuration, the priority of the partitions to set the LIDs under different costs is shown in Figure 10. The darker partition means the higher benefit and the higher priority to set the LID. From Figure 10a, it showed that the east side should be placed with LID before the west side. For 50% cost, the higher priority partitions to set the LID were near the middle and the upstream of the drainage and sewerage system (Figure 10c). When setting a cost of up to 80%, the lower priority LID configuration areas were the upstream mountainous area and the downstream of the major drainage and the sewerage system (Figure 10d).

The Pareto frontier between the cost and the HFR is shown in Figure 11. It also showed that when the cost of the LID was more than 80%, the HRF was almost the same as the 100%. In terms of optimal spatial configuration for the HFR, the priority of the partitions to set the LIDs under different costs is shown in Figure 12. From the 30%–50% cost, the result indicated that the upstream of the major drainage and sewerage system had the higher priority. Therefore, if the cost, the flood peak reduction and the HFR reduction were taken into account at the same time, the highest priority should have been the area in the middle of the major drainage, followed by the upstream of the major drainage, and avoided the area which was allocated to the downstream of the drainage.

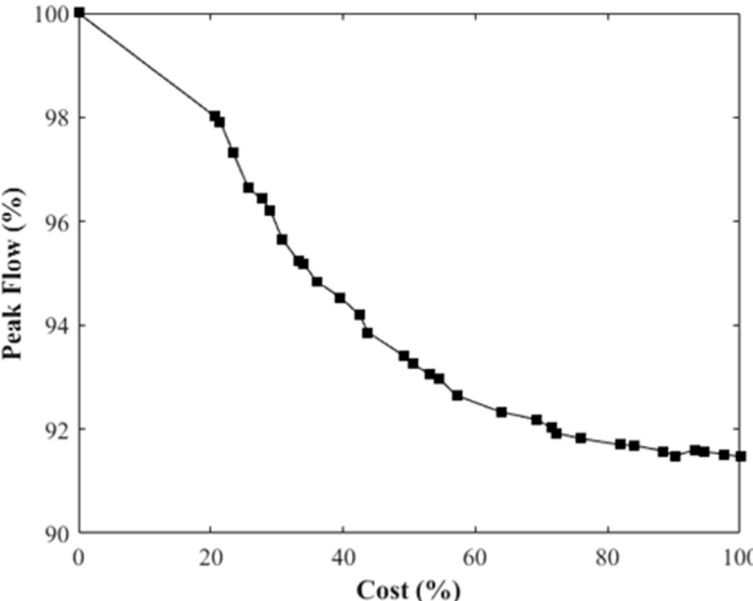

**Figure 9.** The Pareto frontier between the peak flow and the cost.

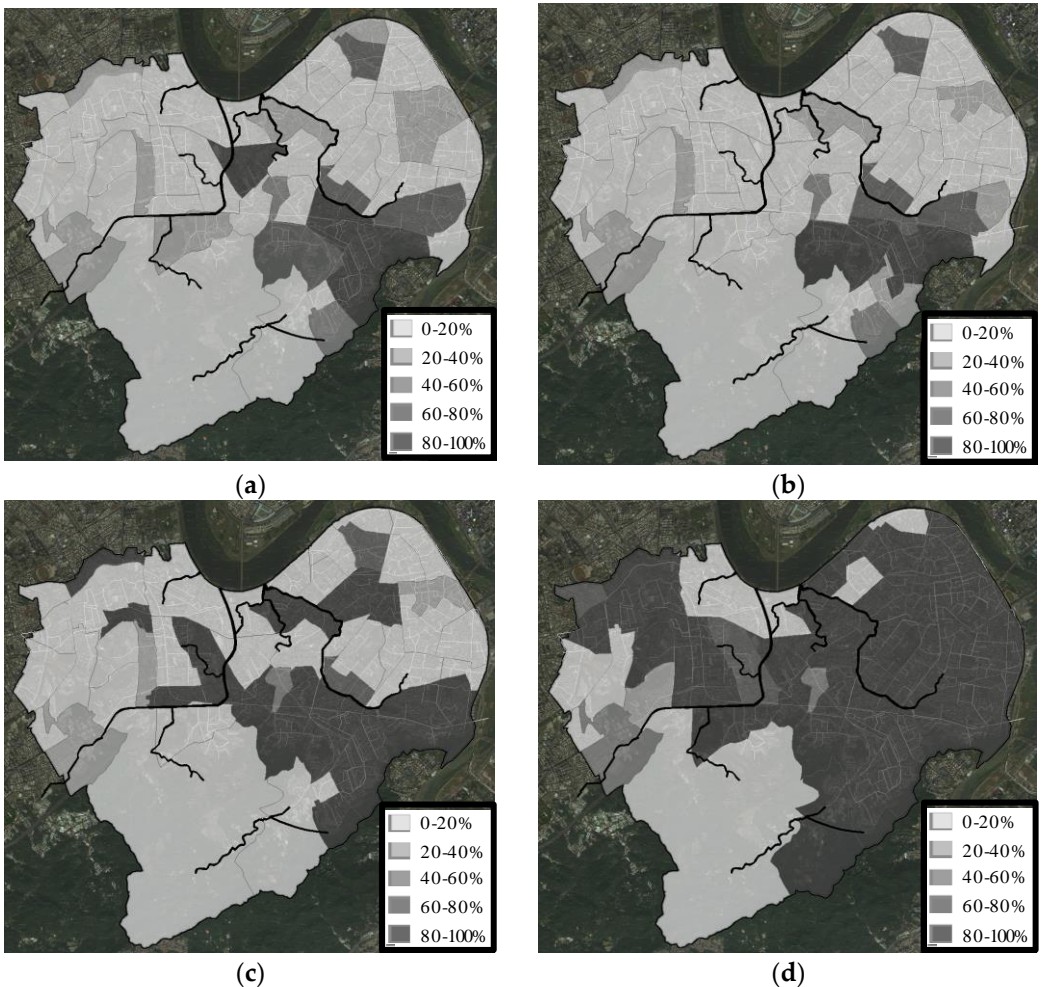

(**a**)                                           (**b**)

(**c**)                                           (**d**)

**Figure 10.** Optimal LID spatial distribution for the flood peak reduction under different LID costs: (**a**) 30% cost, (**b**) 40% cost, (**c**) 50% cost, and (**d**) 80% cost.

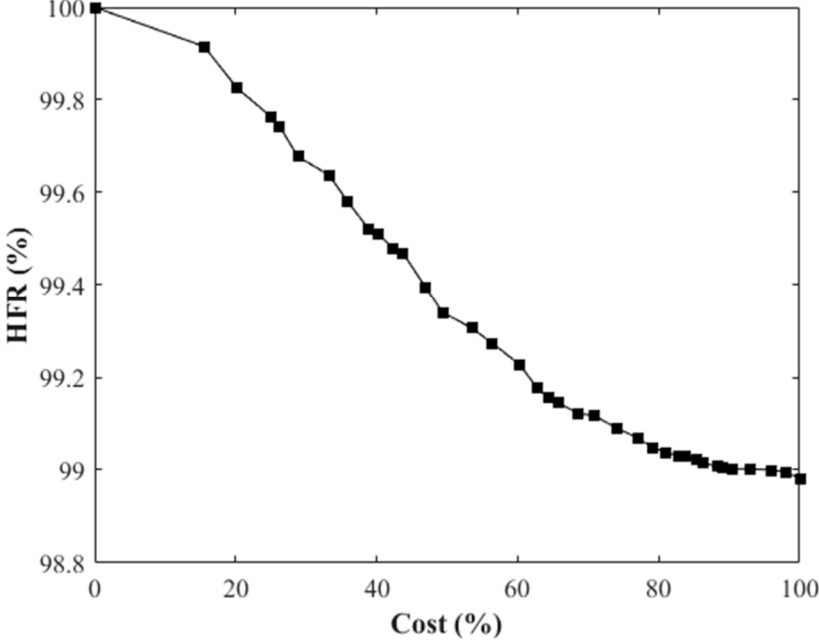

**Figure 11.** The Pareto frontier between the HFR and the cost.

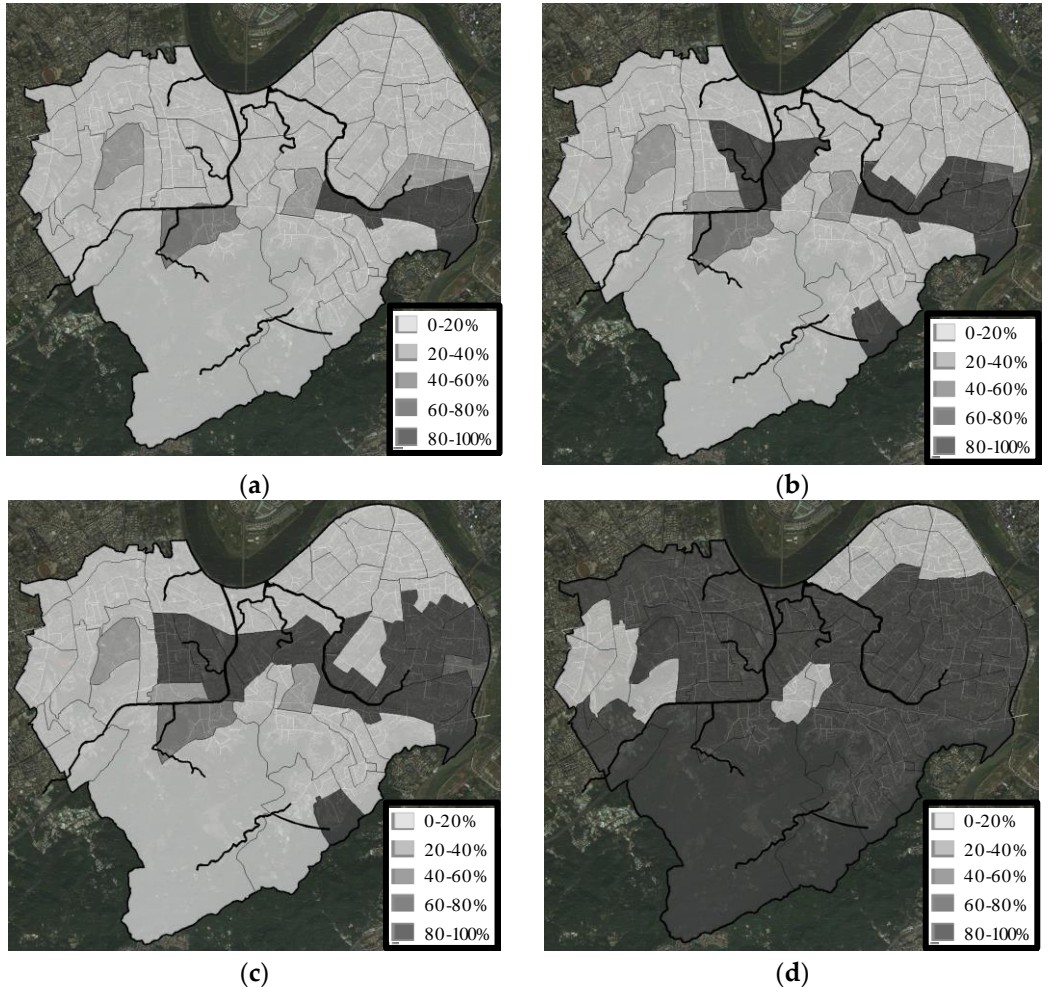

**Figure 12.** Optimal LID spatial distribution for HFR reduction under different LID costs: (**a**) 30% cost, (**b**) 40% cost, (**c**) 50% cost, and (**d**) 80% cost.

### 4.4. Monte Carlo Method

For discussing the optimization of the LID distribution, it may be limited to find the proper area near the upstream or downstream in the major drainage. However, the sewerage system in the city was intricate and it was difficult to define the upstream and downstream areas. It was necessary to configure the LID with other considerations. The aforementioned results showed that although the major drainage two and three can deliver the same discharge, the LID configurations were prone to be placed near the middle and upstream of the drainage two instead of the drainage three. It entailed that there were other factors affecting the optimal setting of LIDs. The Monte Carlo test was conducted to estimate the effect of the land use for the LID configuration in this section. We randomly assigned the land use toward different sub-catchments and the probability of various land use was the same as the original plan (Table 4). The average impermeable rate of each partition was applied to represent the degree of the development. The higher impermeable rate meant the higher degree of the development. In this test the optimal allocation ratio was also estimated to find out the correlation to the degree of the development.

**Table 4.** Probability for the land use in the Monte Carlo test.

| Type of Land Use | Impermeable Rate | Probability |
|---|---|---|
| Woodland | 0 | 0.1 |
| Agricultural land | 18 | 0.05 |
| Land used for park | 19 | 0.1 |
| Land used for school | 60 | 0.1 |
| Land used for agency | 78 | 0.05 |
| Residential district | 84 | 0.2 |
| Land used for public facilities | 87 | 0.05 |
| Industrial district | 90 | 0.05 |
| Commercial district | 97 | 0.1 |
| Land used for road | 100 | 0.2 |

Land use and zoning can be reclassified, therefore the 53 partitions were merged into 23 partitions to reduce the computational time. This experiment produced a total of 2259 sets of samples, so there was a total of 51,957 impermeable rate samples. The distribution percentage of these 51,957 samples is shown in Figure 13. A positive correlation was uncovered between the optimal allocation ratio and the impermeable rate of the partition, no matter if the goal was to reduce the flood peak or the HRF (see Figure 14). If the average impervious rate was below 50%, the performance of LIDs to reduce the flood peak significantly decreased. It meant that the LID setting should be placed in the area where impervious rate was high to obtain the best performance. For the HFR, the result also showed that the highly developed regions needed to set up LIDs to reduce the HFR. The LID distribution result with the Monte Carlo test showed that the LID location in the drainage system was less related to the flood reduction but would affect the HFR reduction. In other words, in terms of the flood reduction, the degree of development could be utilized as the efficient judgment to the LID distribution. Moreover, for urban designers this experiment can be a guideline for the comprehensive planning or the rezoning of the land to include the LID design.

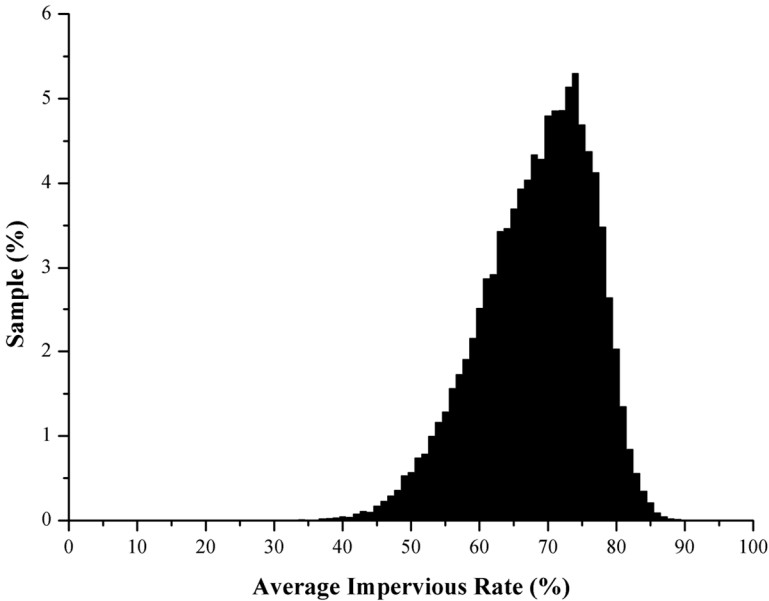

**Figure 13.** The distribution of impervious rates for the Monte Carlo test.

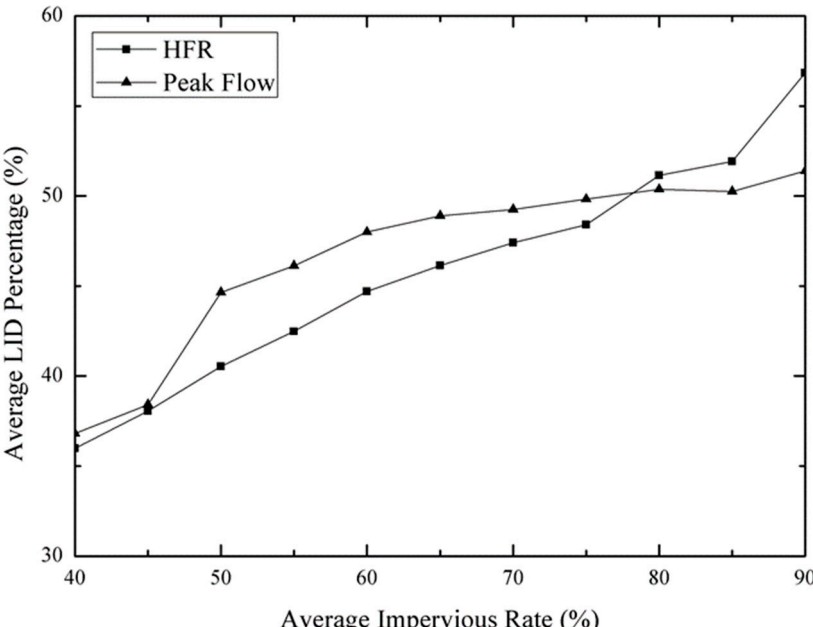

**Figure 14.** The relationship between the optimal allocation ratio and the average impervious rate for the flood peak and the HFR.

## 5. Conclusions

The flood mitigation design with LIDs in a densely populated city was investigated. With the LID suitability design and modeling SWMM, the results showed that the LID units could perform well for small storm events, on the other hand, the detention ponds were more suitable for large storm events in terms of the flood peak reduction. The separation between the small and the large storm events was the 10-year return period. Other than the factor of the flood peak reduction, the HFR was considered as the other factor to evaluate the performance of LIDs and detention ponds through the overall runoff process; and it showed that LIDs could reduce the HFR better than detention ponds under all designed events. We also showed that permeable pavements and green roofs should be first considered if the cost is the main consideration; however, rain barrels should be first chosen if the area is limited.

The results from MOGA showed that the spatial configuration of LIDs would affect the performance of the flood peak reduction and the HFR reduction. In terms of the cost to place the LIDs in the region it showed that the performance of 100% of the region covered with LIDs is similar to 80% of the region covered with LIDs. The effect of the location of LIDs showed that the areas where LIDs were established could reduce the flood peak better than the HFR. With MOGA simulation the priority for placing LIDs were: from the upstream to the downstream for major drainages, but from the downstream to the upstream for minor drainages (such as sewerages) to obtain the favorable reduction of the flood peak and the HFR at the same time. In other words, it was better to place the LIDs closer to the upper reaches of major drainages for the storm control in urban areas.

According to the Monte Carlo test to reproduce the same hydrological and geographical conditions but with different land-use plans, the results showed that the proportion of LIDs set in the region was positively correlated with the average impervious rate. Therefore, for the complex urban drainage system, the LID should be placed in the area where impervious rate is high to obtain the best performance for the flood peak reduction and the HFR reduction. In this study we demonstrated that both the MOGA and Monte Carlo experiments can be conducted to decide the priority of the LID configuration based on the drainage system or the average impervious rate. The results can provide an efficient guideline for LID design in the urban planning.

**Author Contributions:** Conceptualization, H.-Y.L. and H.-C.H.; Methodology, S.-W.L.; Software, S.-W.L.; Validation, C.-C.H., and H.-C.H.; Writing-Original Draft Preparation, C.-C.H., and H.-C.H.; Writing-Review & Editing, C.-C.H. and H.-C.H.; Supervision, H.-C.H.; Project Administration, H.-Y.L. and H.-C.; Funding Acquisition, H.-Y.L. and H.-C.

**Funding:** The study is supported by Ministry of Science and Technology (MOST) of Taiwan, and National Taiwan University under MOST-106-2625-M-002-020-MY2 and MOST-105-2221-E002-063-MY3.

**Acknowledgments:** The outcome for this study were provided by Department of Civil Engineering, National Taiwan University

**Conflicts of Interest:** There is no conflict of interest.

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
