# Peer review of "Evaluation of a Multi-Objective Genetic Algorithm for Low Impact Development in an Overcrowded City"

_water, doi:10.3390/w11102010_

Round 1

Reviewer 1 Report

The paper deals with a somewhat interesting topic. However, the research is either poorly designed or at least presented. Therefore I recommend to reject the manuscript. Here my reasons:

(1)   The English language doesn’t meet the requirements to allow the readers to understand many sections of the text

(2)   Unfortunately, the presentation is very poor and is missing many important issues while using space for topics that are explained elsewhere already. E.g. which model is used for the hydrological assessment, how was it setup, spatial or temporal resolution, input data, calibration?? All important topics for the reader, but not explained. Which LIDs are simulated, how, how are they distributed in the model???

(3)   On the other hand the authors use far too much space to describe all parameters of a GA that seems to be just standard.

(4)   Maybe it’s me or the language, but I didn’t comprehend the HFR concept

(5)   Only a small issue in this case, but the referencing is a mess. References missing in the list, once indexed [1] and then with name and year   

Author Response

Dear reviewer,

We appreciate the valuable comments from you. We addressed each of your comments in the revised manuscript. Our responses to your comments are listed below.

Comments and suggestion:

The paper deals with a somewhat interesting topic. However, the research is either poorly designed or at least presented.

Response: Thank you for the comment. We carefully corrected the missing words, improper use of articles, and inaccurate grammar to clarify this manuscript.

The English language doesn’t meet the requirements to allow the readers to understand many sections of the text.

Response: Thank you. The manuscript was carefully rewrote for many sections to improve the readability. Please check the revised manuscript.

Unfortunately, the presentation is very poor and is missing many important issues while using space for topics that are explained elsewhere already. E.g. which model is used for the hydrological assessment, how was it setup, spatial or temporal resolution, input data, calibration?? All important topics for the reader, but not explained. Which LIDs are simulated, how, how are they distributed in the model?

Response: Thank you for the suggestion. The sections related to SWMM and MOGA were intensively modified. Please check with the revised manuscript. The scenarios used for hydrological assessment in SWMM were calculated with Horner’s equation. The setting for the LID components adopted in this study was explained in section 3.1. The detail of the modification about SWMM setup and optimization approach was in the revised manuscript. Again, thank you for the suggestion. It could make this manuscript much suitable for the readers.

On the other hand, the authors use far too much space to describe all parameters of a GA that seems to be just standard. 

Response: The reviewer is correct. The section for MOGA was modified to reduce the content for all parameters. Please check the revised manuscript in section 3.3.

Maybe it’s me or the language, but I didn’t comprehend the HFR concept.

Response: Thank you. The section for the HFR (section 3.2) was modified to improve the readability.

Only a small issue in this case, but the referencing is a mess. References missing in the list, once indexed [1] and then with name and year 

Response: Thank you. The reviewer is correct. The reference list was re-organized in the revised manuscript. For “Line 26”, it was modified as “According to the report from the United Nations, 55% of the 7 billion population is concentrated in cities, and urban drainage problems are increasing as a result of the high density population [1].”

Reviewer 2 Report

Very good manuscript, with an interesting approach.

I have some comments on the manuscripts. listed below:

- English needs a careful review.

Line 15: and then the detention ponds were conducted to estimate the improvement of urban 15 resilience
Review the English and be careful with overuse of the term Resilience.

Line 18: 'and to find out the location 18 sequence that should be first configured in the densely-dense development cities."
Not clear what you mean with 'sequence that should be first configured'.

Line 42: "Compare various models to simulate LID [6], it shows" ..review

Line 56: 'Liu et 2016 developed new LID best algorithms in combination with the self-56 developed model (L-THIA-LID 2.1 combined with AMALGAM) [15]."
Please clarify and re-write

Line 64: Giacomoni et al.(2009, 2012) .. fix the reference

Fig 1: would be useful for the reader to show were in the country the region is located.
Maps also need scale and units

Fig 11 and 13: could be presented in colours for easier reading.

Author Response

Dear reviewer,

We appreciate the valuable comments from you. We have addressed each of your comments in the revised manuscript. Our responses to your comments are listed below.

Comments and suggestion:

Abstract

Very good manuscript, with an interesting approach.

Response: Thank you for the comments.

Line 15: and then the detention ponds were conducted to estimate the improvement of urban resilience. Review the English and be careful with overuse of the term Resilience.

Response: Thank you for the suggestion. The sentence was modified to “and then the detention ponds were conducted to estimate the improvement of urban resilience.”. Please see “Line 15-16”.

Line 18: 'and to find out the location sequence that should be first configured in the densely-dense development cities." Not clear what you mean with 'sequence that should be first configured'.

Response: We are agreed with the reviewer. This sentence was ambiguous and then was modified to “The multi-objective genetic algorithm (MOGA) was therefore conducted to optimize the spatial configuration of LIDs under different budget scenarios, and to decide the priority of locations for the LID configuration”. Please see “Line 17-19”.

Line 42: "Compare various models to simulate LID [6], it shows" ..review

Response: Thank you for the suggestion. The sentence was modified to “For numerical models for LID simulation in urban areas, it shows that EPA’s Storm Water Management Model (SWMM) works best under the considerations of time, space, accuracy and calculation efficiency [6-8]. “. Please see “Line 43-45”.

Line 56: 'Liu et 2016 developed new LID best algorithms in combination with the self developed model (L-THIA-LID 2.1 combined with AMALGAM) [15]." Please clarify and re-write.

Response: The authors are agreed with the reviewer. The sentence was re-wrote as Liu et al.(2016)developed the new model which was named as Long-Term Hydrologic Impact Assessment-Low Impact Development 2.1 (L-THIA-LID 2.1) to estimate the optimization of LID configurations to mitigate the impact of urbanization and climate change [15].” . Please see “Line 57-59”.

Line 64: Giacomoni et al.(2009, 2012) .. fix the reference.

Response: Thank you. The reference was fixed and it was changed to “Giacomoni et al. (2009, 2011) proposed ……”. Please see “Line 66”.

Fig 1: would be useful for the reader to show were in the country the region is located. Maps also need scale and units.

Response: Thank you. The authors are agreed with this suggestion to clarify the map. The modified figure was shown in the manuscript. Please see “Line 102”.

Fig 11 and 13: could be presented in colours for easier reading.

Response: Thank you for the suggestion. The variation of the grayscale was adopted in Fig 11 and 13 to show the tendency of the optimal locations of LID configuration. We will explain in the manuscript for the readers.

Round 2

Reviewer 1 Report

The revised version shows clear improvements. The methods section is now much shorter and at the same time clearer and rather concise. However, some things are still to be improved (even though I mentioned some of these in the first review already). I would kindly ask you to carefully consider my comments and reply to them:

(1) the reference style is still not concise. Once you use numbers in the text and once names and years. Correct me if I'm wrong, but that's not how it should be. Please thoroughly check your references, use a concise style, and check whether all of them are in the reference list. I marked some in the attached file.

(2) the language still needs improvement. Please have the document language checked. I marked some issues, but most likely not all.

(3) you replied you modified section 3.2 on the HFR concept. First, I don't see any changes and second, I obviously still think it's not clearly explained.

(4) you should discuss your findings with other research. You don't have a single reference in the discussion. So, please reflect your findings with the findings of other research. That's a main part of publishing results.

(5) the explanation of the concept is now much improved. Please still state somewhere (e.g. in a table) how much area (e.g. as a fraction) is occupied by different LIDs for each scenario.

(6) you still don't mention anywhere how the initial model was setup? how did you calibrate or validate? Did you do some sensitivity check? Where are model parameters coming from (e.g. for the subcatchments or the drainage network)?

Author Response

Dear reviewer,

We appreciate the valuable comments from you. We have addressed each of your comments in the revised manuscript. Our responses to your comments are listed below.

Comments and suggestion:

The revised version shows clear improvements. The methods section is now much shorter and at the same time clearer and rather concise. However, some things are still to be improved (even though I mentioned some of these in the first review already). I would kindly ask you to carefully consider my comments and reply to them:

Response: Thank you for the comments. We modified the manuscript based on your comments in this version.

The reference style is still not concise. Once you use numbers in the text and once names and years. Correct me if I'm wrong, but that's not how it should be. Please thoroughly check your references, use a concise style, and check whether all of them are in the reference list. I marked some in the attached file.

Response: Thank you for the comments. We checked the manuscript and modified the corresponding sentences, for example, the original sentence is “Ho (2014) pointed out that the bioretention cell is 46 superior to the permeable pavement, and did not recommended to use bioswale [9]”. The new sentence is “Ho [9] pointed out that the bioretention cell is 46 superior to the permeable pavement, and did not recommended to use bioswale”. Please see “Line 46, 47, 49, 54, 56, 57, 59, 60, 65, 181, 209, 213, 269” where we modified all the references.

The language still needs improvement. Please have the document language checked. I marked some issues, but most likely not all.

Response: The reviewer is correct. The language was improved based on the reviewer’s suggestions. Please see the revised manuscript. Thank you.

You replied you modified section 3.2 on the HFR concept. First, I don't see any changes and second, I obviously still think it's not clearly explained.

Response: Thank you for the correction. The section related to HFR was modified in the revised manuscript. The HFR is basically the product of the amount of land that was inundated, and the duration of the time that each unit of land was inundated. If we plot the inundated land at each time step during the event, the area under this curve is termed as HFP. Please see more description in section 3.2.

You should discuss your findings with other research. You don't have a single reference in the discussion. So, please reflect your findings with the findings of other research. That's a main part of publishing results.

Response: Thank you. We added other findings which showing the efficiency of LID components would depend on the return period and the duration of rainfall if the flood peak reduction is the purpose. These results are similar to our Figures 6 and 7. Please see Line 269.

The explanation of the concept is now much improved. Please still state somewhere (e.g. in a table) how much area (e.g. as a fraction) is occupied by different LIDs for each scenario.

Response: Thank you. The portion of each land use type in the study area was mentioned in Table 2. The area occupied by LIDs was calculated by area of each subcatchment and the reduction factor. There were 6,639 subcatchments employed in SWMM simulation. Even though we decreased into 53 partitions while doing optimization, it might be not easy to show the percentage of occupied LID in each partition under different scenarios. Therefore, we adopted Figs 10 and 11 to show the priority of the place to install LIDs under different scenarios. The figures can also inform the trend to install LIDs in this study area. The setting of parameter for the simulation was added in section 3.1 for more information.

You still don't mention anywhere how the initial model was setup? how did you calibrate or validate? Did you do some sensitivity check? Where are model parameters coming from (e.g. for the subcatchments or the drainage network)?

Response: Thank you. The initial setup for the model was added in section 3.1. The parameters applied for subcatchments was surveyed by the New Taipei City government, and the data are available online (http://wrs.ntpc.gov.tw/rwweb/)

The model validation was discussed in section 4 (please see “Line 207-215”). The surveyed inundation areas during typhoon events were adopted to compare with our simulation results. The flooding areas were calculated with the in-house 2D model (labeled as NTU-2DFIM). The interaction between this 2D model and SWMM was published by Chen et. al (2003). We used Typhoons Herb and Nari as our verified cased because the inundation area would only be surveyed for extreme events. The comparisons are shown in Fig a and Fig b. The accuracy was 96.3% and 89.5% for two events. From the aforementioned result, we demonstrated the setting for SWMM is appropriate and SWMM is robust. Moreover, the purpose of this manuscript is trying to show that MOGA with SWMM simulation result can be conducted as optimization method in the large area (24.22 km2). Moreover, this method can also indicate the priority for the places to install LID in the large area if the cost is limited.

Fig a. and b are in the attached file (water-554632_20190823_Draft; Page 3 of 3)

Round 3

Reviewer 1 Report

Concerning the language deficits, the authors seem to not take sufficient effort for improvement. The correction of the language is not the reviewers task. I gave some examples for grammar and wording mistakes. But I did not correct the entire document (as I wrote).

So, again, I kindly ask you to have the manuscript language checked to improve the presentation.

I asked you to extend the discussion. I don't think the implementation of one reference is sufficiently addressing my request.

Author Response

Dear reviewer,

We appreciate the valuable comments from you. Our responses to your comments are listed below.

Comments and suggestion:

Concerning the language deficits, the authors seem to not take sufficient effort for improvement. The correction of the language is not the reviewers task. I gave some examples for grammar and wording mistakes. But I did not correct the entire document (as I wrote). So, again, I kindly ask you to have the manuscript language checked to improve the presentation.

Response: The reviewer is correct. We carefully corrected the missing words, inaccurate grammar, and the improper use of the articles in this revised manuscript. The presentation about the HFR, the SWMM modeling set-up, and other parameters conducted in the simulation was re-wrote to enhance the readers’ understanding. Please check the highlight sentences and words in this revised manuscript. The calibration of SWMM model was explained in the response document after the references section in the revised manuscript.

I asked you to extend the discussion. I don't think the implementation of one reference is sufficiently addressing my request.

Response: Thank you for the comments. We added four references related the management of LID allocation in the small area through numerical modeling, and one paper discussed about the numerical simulation for the storm control in urban area. The discussion was inserted in section 4 and 4.1. The list of these references is shown below:

Chen, S.H.; Hsu, M.H.; Chen, T.S. Simulation for interactions between storm sewer and overland flows, New Pipeline Technologies, Security, and Safety. 2003, 437–446. Bai, Y.; Zhao, N.; Zhang, R.; Zeng, X. Storm Water Management of Low Impact Development in Urban Areas Based on SWMM. Water. 2019, 11(1), 33; doi:10.3390/w11010033. Pereira Souza, F.; Leite Costa, M. E.; Koide, S. Hydrological Modelling and Evaluation of Detention Ponds to Improve Urban Drainage System and Water Quality. Water. 2019, 11(8), 1547; doi:10.3390/w11081547 Rezaei, A. R.; Ismail, Z.; Niksokhan, M. H.; Dayarian, M. A.; Ramli, A. H.; Shirazi, S. M. A Quantity–Quality Model to Assess the Effects of Source Control Stormwater Management on Hydrology and Water Quality at the Catchment Scale. Water. 2019, 11(7), 1415; doi:10.3390/w11071415. Liang C.Y.; You, J.Y.; Lee, H.Y. Investigating the effectiveness and optimal spatial arrangement of low-impact development facilities. Journal of Hydrology. 2019, 577, 124008.
